# Magnetically brightened dark electron-phonon bound states in a van der Waals antiferromagnet

Emre Ergeçen[1,5], Batyr Ilyas [1,5], Dan Mao[1], Hoi Chun Po[1,2], Mehmet Burak Yilmaz[1], Junghyun Kim[3,4], Je-Geun Park [3,4], T. Senthil[1] & Nuh Gedik [1✉]

In van der Waals (vdW) materials, strong coupling between different degrees of freedom can hybridize elementary excitations into bound states with mixed character[1–3]. Correctly identifying the nature and composition of these bound states is key to understanding their ground state properties and excitation spectra[4,5]. Here, we use ultrafast spectroscopy to reveal bound states of $d$-orbitals and phonons in 2D vdW antiferromagnet $NiPS_3$. These bound states manifest themselves through equally spaced phonon replicas in frequency domain. These states are optically dark above the Néel temperature and become accessible with magnetic order. By launching this phonon and spectrally tracking its amplitude, we establish the electronic origin of bound states as localized $d$–$d$ excitations. Our data directly yield electron-phonon coupling strength which exceeds the highest known value in 2D systems[6]. These results demonstrate $NiPS_3$ as a platform to study strong interactions between spins, orbitals and lattice, and open pathways to coherent control of 2D magnets.

[1] Department of Physics, Massachusetts Institute of Technology, Cambridge 02139 MA, USA. [2] Department of Physics, Hong Kong Univesity of Science and Technology, Clear Water Bay, Hong Kong 999077, China. [3] Center for Quantum Materials, Seoul National University, Seoul 08826, Republic of Korea. [4] Department of Physics and Astronomy and Institute of Applied Physics, Seoul National University, Seoul 08826, Republic of Korea. [5] These authors contributed equally: Emre Ergeçen, Batyr Ilyas. ✉email: gedik@mit.edu

nteraction between electrons and lattice is at the core of many physical phenomena observed in 2D vdW materials, such as interlayer exciton–phonon states[7] and enhanced superconductivity[8]. One of the hallmarks of strong electron–phonon coupling is electron–phonon bound states, a product of coherent coupling between electronic and vibrational levels. In experiments such as photoemission[9] and optical spectroscopy[6], these hybrid excitations in two dimensional vdW materials give rise to spectral replicas equally spaced by a phonon frequency. These states can be used to quantify electron–phonon coupling without relying on any physical models.

In vdW magnets, lattice plays a crucial role in determining the magnetic ground state properties. For example, changes in stacking configuration drive the prototypical bulk vdW ferromagnet $CrI_3$ to an antiferromagnet in few layer limit[2]. The spin–lattice coupling also manifests itself in the vibrational modes of vdW magnets[10] as an energy renormalization below the magnetic ordering temperature. In addition to the strong coupling to the lattice, magnetic ordering leads to the formation of new electronic states, that are exploited as a proxy for antiferromagnetic order in 2D limit, such as $d$–$d$ transitions in transition metals[11] and spin–orbit entangled excitons[12,13].

This strong coupling between different degrees of freedom in vdW magnets can lead to novel bound states with mixed character. These hybrid excitations could pave the way to achieve new functionalities for manipulating optical and electronic properties. Despite this great promise, their true potential has not yet been realized in vdW magnets. The main challenge is to identify and characterize these excitations. Deciphering the components of a bound state can be challenging in materials with multiple electronic excitations and collective modes. This requires spectral tools that can differentiate between various bosonic and fermionic components partaking in the bound state formation.

In this letter, we use coherent ultrafast spectroscopy with energy and time resolution to reveal bound states of localized $d$-orbitals and a phonon in the 2D antiferromagnetic vdW insulator $NiPS_3$. These bound states are optically dark above the Néel temperature ($T_N$) and become optically accessible below $T_N$. They manifest themselves as equally spaced 10 replica peaks in transient absorption spectra with an energy spacing that corresponds to the 7.5 THz (253 cm$^{-1}$) $A_{1g}$ Raman active phonon mode. In order to pin down the electronic origin of these spectral progressions, we use energy-resolved coherent phonon spectroscopy to launch this phonon mode coherently and watch it both in time and frequency domains. Our data unequivocally shows that the phonon replicas originate from localized $d$–$d$ electronic transitions.

## Results

$NiPS_3$ is a member of transition metal thiophosphate (MPX$_3$, M: Fe, Mn, Ni, and X: S, Se) layered crystal family[14]. In $NiPS_3$, the arrangement of nickel atoms forms a honeycomb lattice (Fig. 1c). Below the Néel temperature (~150 K), spins localized at nickel sites align in-plane, ferromagnetically along zigzag chains and antiferromagnetically between the adjacent chains. There is also a slight out-of-plane canting of spin orientations[15]. At low temperatures, the system develops several spectral features in the near-infrared and visible spectrum, such as spin–orbit-entangled excitons and on-site $d$–$d$ transitions as observed in linear absorption measurements[11,16]. These are in close proximity to the band edge absorption (1.8 eV).

To investigate the equilibrium and non-equilibrium optical properties of $NiPS_3$, we employ a broadband transient absorption spectroscopy with a high dynamic range. In this experimental

scheme, high intensity (4.3 mJ/cm$^2$) ultrashort pulses with an energy of 1.88 eV efficiently generate electron and hole pairs and heat up the magnetic and electronic systems. The demagnetizing effect of high-intensity optical pulses has been extensively studied in magnets[17–19] and is known to bleach any linear or nonlinear spectral responses pertinent to the magnetic ordering. Subsequent to photoexcitation, a broadband probe pulse with energy spanning from 1.4 to 2.0 eV measures the changes in the reflectivity spectrum. Transient reflectivity changes probed 2 ps after the pump are shown in Fig. 1b. Below 1.53 eV, the spectrum shows two sharp absorption lines (Peaks I and II in Fig. 1b), between 1.45 and 1.50 eV, which have been previously identified as a spin–orbit-entangled exciton and a magnon sideband[11]. As shown in Fig. 2a, they become visible only below the antiferromagnetic ordering temperature (~150 K), pinning down their magnetic origin. Above 1.53 eV, the spectrum is dominated by a broad peak that has been assigned to localized electronic transition among $d$-orbitals[16]. The striking observation is the fine oscillations on top of the transition peak near 1.7 eV (Fig. 1b), reminiscent of spectral replicas. The temperature-dependent data (Fig. 2) shows that the onset of these spectral structures coincides with the Néel temperature. The Fourier transform of spectral replicas exhibits a broad distribution between 25 and 35 meV and peaks around 28.5 meV (Fig. 1b, inset).

Spectral replicas are ubiquitous in molecular systems with strong vibronic coupling[20], in semiconductors with localized excitations[6,21–23], and in engineered quantum systems[24–26]. In the case of $NiPS_3$, one of the potential explanations of regularly spaced spectral replicas is the dressing of an electronic excitation, such as excitons or $d$–$d$ transitions, with a bosonic field, such as phonon or magnon. We rule out the coupling between electronic excitations and magnons as a potential explanation for spectral replicas, because the magnon modes soften as they approach the critical temperature and the energy spacing of replicas does not change with increasing temperature (Supplementary Fig. S5). As corroborated by Raman spectroscopy results[10,27], the energy spacing between replicas matches well with an $A_{1g}$ phonon mode, which is responsible for out-of-plane even-symmetry vibrations of sulfur atoms (Fig. 1c). This observation indicates that the replicas we observe originate from 7.5 THz (253 cm$^{-1}$) $A_{1g}$ phonon mode.

Although we identified the bosonic field involved in the hybrid state formation as phonons, its electronic constituent still remains unresolved in the light of our transient absorption measurements. The coupling of this phonon mode to either $d$–$d$ transitions or spin–orbit-entangled excitons can explain the emergence of replicas. Since these spectral responses pertaining to both of these electronic elements onset at the Néel temperature[16], the temperature dependence can not distinguish between these two scenarios and pinpoint the electronic origin of replicas.

To address this question, we seek to determine the phonon coupling with each spectral region using energy-resolved coherent phonon spectroscopy (see "Methods" section). Unlike the broadband transient absorption spectroscopy experiment with high intensity (~4.33 mJ/cm$^2$) and longer pulse duration (~200 fs), in this experimental scheme a low intensity (~1 μJ/cm$^2$) and ultrashort (~25 fs) pump pulse impulsively or displacively drives the phonon modes with $A_{1g}$ and $E_g$ symmetries without perturbing the underlying magnetic order. Subsequent to the pump excitation, a broadband and ultrashort probe pulse tracks these phonon oscillations as a function of both time and wavelength. The energy integrated coherent phonon spectroscopy traces (Fig. 3a) exhibit an oscillatory part, which represents coherently excited phonon modes, overlaid on an incoherent electronic relaxation. The Fourier transform of these oscillations is given in the inset of Fig. 3a. We observe two $A_{1g}$ phonons at 7.5 THz (253 cm$^{-1}$) and

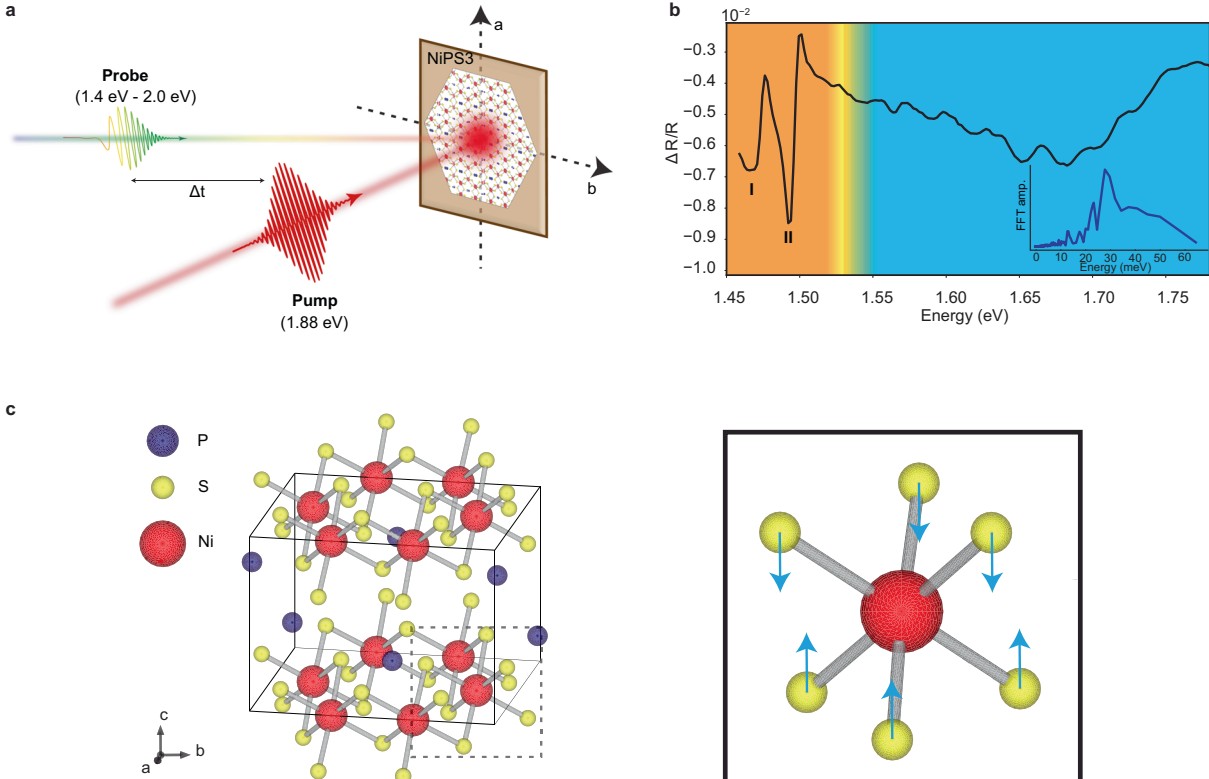

**Fig. 1 Phonon replicas observed in the transient absorption spectrum of NiPS₃. a** Schematic of the broadband transient absorption spectroscopy setup. A pulse with energy (1.88 eV) larger than the bandgap excites the system, and a subsequent probe pulse with white light continuum measures the transient reflectivity in a broad spectral region, with 1.5 meV spectral resolution. **b** Spectral dependence of transient reflectivity measured at 10K, 2 ps after the pump. Previously reported coherent many-body spin–orbit-entangled excitons (orange region) and a broad $d$–$d$ transition (blue region) can clearly be seen. Two sharp features are labeled as Peak I and Peak II which have been previously identified as a spin–orbit-entangled exciton and a magnon sideband[11]. The $d$–$d$ peak is dressed by phonons, with at least 9 replicas visible, indicating a strong coupling between them. The inset shows the Fourier transform of the extracted spectral oscillations, with a mode peaking at 28.5 meV, which is close to the 7.5 THz $A_{1g}$ phonon energy. **c** Crystal structure of NiPS₃. Octahedron as shown in the solid square, formed by a nickel (red) and six sulfur (yellow) atoms. Phosphorus atoms are shown in dark blue. Arrows indicate the $A_{1g}$ phononic displacements for the 7.5 THz $A_{1g}$ mode.

11.5 THz (384 cm⁻¹) and one $E_g$ phonon at 5.2 THz (173 cm⁻¹). The energy of the dominant mode (7.5 THz) matches well with the energy spacing of the replicas.

To examine this phonon mode's coupling strength to $d$–$d$ transitions and spin–orbit-entangled excitons, we compare the oscillation amplitudes obtained from different spectral locations by spectrally separating the broadband probe pulse with the help of a filter (Fig. 3a) or a monochromator (Fig. 3b). We first use a low pass filter with a cutoff at 1.53 eV to coarsely focus on the dynamics of the excitons and to suppress the response of $d$–$d$ transitions. Transient reflectivity traces obtained with and without the low pass filter as shown in Fig. 3a, exhibit a salient feature. Although most of the incoherent response is localized below 1.53 eV, where the excitons reside, the $A_{1g}$ phonon mode oscillations are absent in this spectral region, implying their negligible coupling to spin–orbit-entangled excitons.

The finer spectral dependence of the phonon amplitude obtained with a monochromator is shown in Fig. 3b. Coherent phonon oscillations start to build up above 1.53 eV, and are not present in the spin–orbit-entangled exciton region. Therefore, this observation indicates that spin–orbit-entangled excitons do not play a role in the replica formation. Given that the coherent phonon oscillations are only observed in the spectral region of $d$–$d$ transitions, we conclude that the replica peaks emerge as a result of hybridization between localized $d$–$d$ levels and a Raman active optical phonon.

## Discussion

The energy level splitting of a $d$–$d$ transition is proportional to the distance between the transition metal ion and the ligands surrounding it. In the case of NiPS₃, as shown in Fig. 1c, a 7.5 THz phonon with $A_{1g}$ symmetry modulates interatomic octahedral distances and consequently the energy splittings between $d$-levels. Therefore, the following effective Hamiltonian of this minimal model captures the dynamics of our system

$$H = \hbar(\omega_{dd} + M(\hat{a} + \hat{a}^\dagger))\hat{\sigma}^\dagger\hat{\sigma} + \hbar\omega_{ph}\hat{a}^\dagger\hat{a} \qquad (1)$$

where $\hat{\sigma}$ and $\hat{\sigma}^\dagger$ denote annihilation and creation operators of the fermionic $d$–$d$ transitions, and $\hat{a}$, $\hat{a}^\dagger$ are operators of the 7.5 THz $A_{1g}$ phonon field. The first term describes the $d$–$d$ electronic transition with energy $\hbar\omega_{dd}$ as a two-level system coupled to a phonon field with a coupling constant of $M$. The second term is the total energy of non-interacting phonons of energy $\hbar\omega_{ph}$. The coupling constant can be written as $M = \sqrt{g}\omega_{ph}$[28], where $g$ is called the dimensionless Huang–Rhys factor, which quantifies electron–phonon coupling strength. This Hamiltonian is exactly diagonalizable with a unitary transformation[28] and the spectral function $A(\omega)$ has a form of Poisson distribution:

$$A(\omega) = 2\pi e^{-g}\sum_{n=0}^{\infty}\frac{g^n}{n!}L(\omega - \omega_{dd} + \Delta - \omega_{ph}n) \qquad (2)$$

where $L$ is a function corresponds to the lineshape of undressed $d$–$d$ transition, $n$ is the number of phonons and $\Delta$ is the

renormalization energy of $d$–$d$ transition due to hybridization with phonons, which is equal to $g\omega_{ph}$. To extract the electronic transition energy and the $g$ factor, we fit our transient absorption traces with this spectral function (Fig. 4). The $g$ factor as extracted from the fit is estimated to be ~10. This exceptionally high value indicates a strong coupling between the d-levels and vibrational modes that modulate the octahedral distances.

These spectral features start to become apparent at a temperature range, in which the thermal occupation for a 7.5 THz (253 cm$^{-1}$)

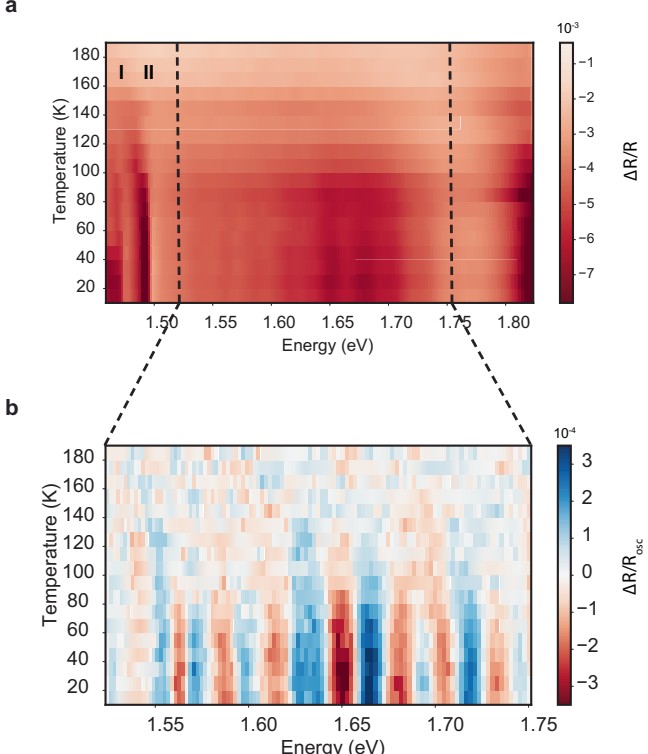

**Fig. 2 Temperature dependence of phonon replicas. a** Temperature dependence of the spin–orbit-entangled excitons and spectral oscillations. Exciton states become visible below the Néel temperature, as reported previously. Phonon-dressed $d$–$d$ excitations also follow the Néel order. **b** Enlarged view of the $d$–$d$ transition region. The background is subtracted using a fifth-order polynomial fit.

phonon mode is negligible. The amplitudes of replicas get enhanced with decreasing temperature, demonstrating the coherence formed between $d$–$d$ excitations and phonons. Additionally, the extracted amplitude of each phonon replica peak follows a Poisson distribution, rather than a Bose–Einstein distribution that describes a thermal state[25]. Therefore, the spectral replicas we observe correspond to a coherent superposition between the $d$–$d$ electronic transition and different phonon number states, not a thermal ensemble.

Generally, transitions between $d$-orbitals are optically forbidden in a centrosymmetric environment because of dipole selection rules. These transitions can still be made optically accessible by perturbations that transiently or permanently break local inversion symmetry at nickel sites. Although stacking faults and lattice defects can break inversion symmetry both globally and locally, it is unlikely that they result in the appearance of $d$–$d$ transitions concomitant with the magnetic order. On the other hand, in the case of transiently broken inversion symmetry by odd-symmetry phonons, the spectral weight of $d$–$d$ transitions is expected to increase with increasing temperature[20], due to the thermal occupation of phonon modes. However, in NiPS$_3$, the $d$–$d$ transition becomes visible only below the Néel temperature, as shown in Fig. 2, ruling out a phonon-driven inversion symmetry breaking scenario. This fact, as also seen in linear absorption measurements[16], hints at a mechanism that breaks local inversion symmetry at nickel sites ensuing from magnetic order. In Fig. 5, we illustrate the local environment of nickel sites both above and below the ordering temperature. At high temperatures, all of the sulfur atoms are located between indistinguishable nickel ions without a particular spin orientation, preserving the local inversion symmetry. However, at low temperatures, the sulfur atoms between two anti-aligned spins (indicated in orange) become distinct from those adjoining two aligned nickel spins (indicated in yellow), due to the charge transfer nature of NiPS$_3$[11]. As the exchange interaction is mediated by sulfur atoms, their spin and charge configurations are affected by surrounding nickel spin orientations, leading to distinguishable ligands. Below Néel temperature, this dissimilarity between sulfur atoms breaks local inversion symmetry, as the inversion operation centered at a nickel ion links two sulfur atoms of different kinds (Fig. 5). Therefore, even though the antiferromagnetic order preserves global inversion symmetry[29] in NiPS$_3$ (Supplementary Fig. S8), the local inversion symmetry at nickel sites is broken, rendering electronic transitions among d-levels dipole allowed.

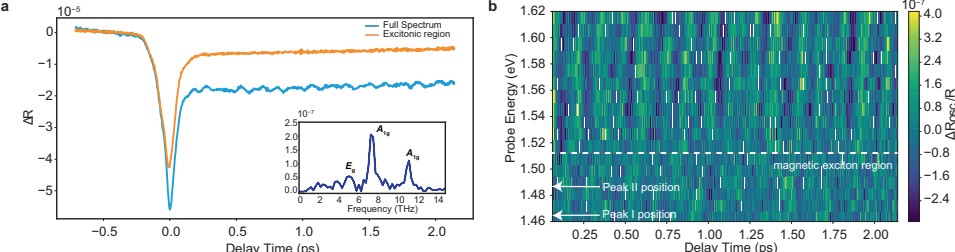

**Fig. 3 Phonon modulates spectral region near $d$–$d$ transition. a** Coherent phonon spectroscopy traces were obtained below the Néel temperature (77 K), with ~25 fs temporal resolution. The blue trace represents the energy integrated coherent phonon spectroscopy trace. The oscillatory part of the energy integrated transient reflectivity was extracted after subtracting an exponential background from the raw data. Energy integrated coherent phonon spectroscopy reveals three distinct high-frequency optical phonons at 5.2, 7.5, and 11.5 THz. The Fourier amplitudes of the oscillations are shown in the inset. The orange line represents the data obtained with a long pass filter (cut-off energy: 1.53 eV) that excludes the $d$–$d$ level transition. This spectrally separates the spin–orbit-entangled exciton from $d$–$d$ levels. The 7.5 THz phonon mode modulates the $d$–$d$ transition region, whereas the region with excitons is not being modulated by this mode. **b** Energy-resolved coherent phonon spectroscopy. For finer energy resolution, the probe beam was sent to a monochromator and coherent phonon oscillations in the energy region between 1.46 and 1.62 eV, was measured. The data clearly shows that the modulations are mainly present in the spectral region near the $d$–$d$ transition.

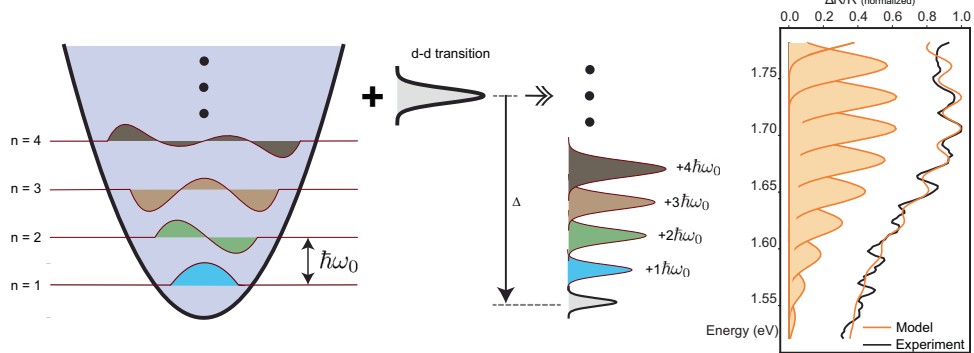

**Fig. 4 Extracting the Huang–Rhys factor.** On the left, phonon eigenstates with equal spacings are shown. Coupling between the *d*–*d* transition and phonon degrees of freedom gives rise to a broad peak in absorption with an oscillatory feature on top as well as an energy renormalization denoted by Δ. This model matches well with the experimental data. We fit the absorption spectrum with a minimal model which consists of a sum of linewidth functions (Gaussian or Lorentzian) weighted by a Poisson distribution. The free parameter of this distribution corresponds to the dimensionless Huang–Rhys factor *g*.

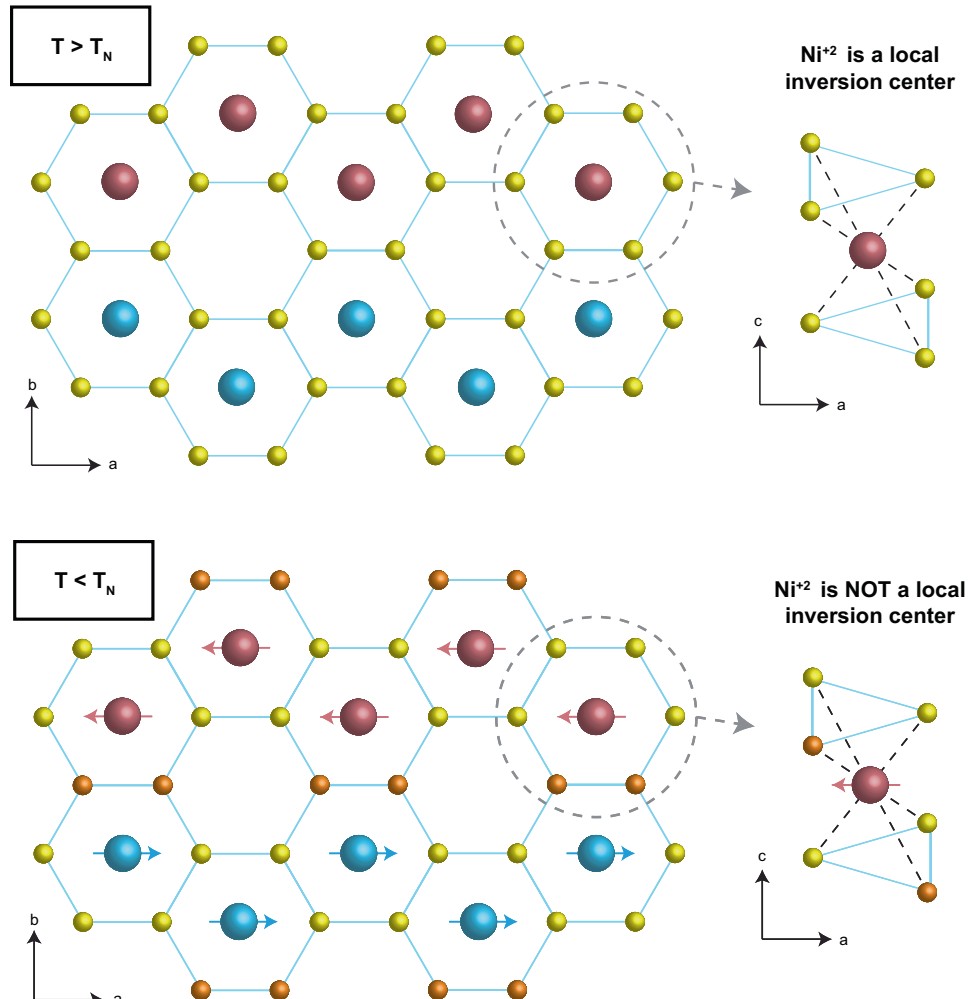

**Fig. 5 Magnetism breaks local inversion symmetry enabling *d*–*d* transition.** With the onset of Néel order, the local environment around nickel sites breaks the local inversion symmetry and makes the electric-dipole forbidden *d*–*d* transition visible in the absorption spectrum. For $T > T_N$, all sulfur atoms (yellow) are equivalent, and hence nickel site is a local inversion center. For $T < T_N$, nickel atoms are no longer local inversion centers, as the magnetic order creates two sets of distinguishable sulfur atoms: the ones between the opposite spins (orange) and the ones between the parallel spins (yellow). These dissimilar sulfur atoms break local inversion at nickel sites, as shown on the right.

In summary, we observe magnetically brightened dark electron–phonon-bound states of a localized $d$–$d$ transition and $A_{1g}$ phonon mode in NiPS$_3$. Using ultrafast methods that quench the magnetic order and coherently launch phonon modes, we resolve the fine structure of spectral replicas and conclusively determine their bosonic and fermionic components. The extracted coupling strength among $d$-orbitals and phonons is exceptionally high ($g \sim 10$) and exceeds the known highest value in vdW materials, CrI$_3$[6] ($g = 1.5$), by nearly an order of magnitude. Our study shows that in NiPS$_3$ the well-defined in-gap electronic states are not only strongly coupled to photons and magnons[11–13], but also to phonons. These spectrally separated in-gap states suggest the possibility of using different optical drives to achieve effective magnon–phonon coupling[30]. Furthermore, in transition metal thiophosphates, crystal field splitting determines the magnetic anisotropy, and has a central role on the type of magnetic order[31,32]. Hence, our findings show that metastable magnetic states could be realized in NiPS$_3$ through the efficient control of crystal field splittings. This strong coupling is amenable to controlling $d$–$d$ electronic transitions either transiently through nonlinear phonon driving[33] or in equilibrium through strain or pressure. Our results also indicate that vdW magnets could be ideal platforms to engineer novel phases of matter by using phonon-driven Floquet states[34,35].

*Note added:* During the completion of this work, we became aware of a complementary work[13] using wavelength resolved birefringence to resolve the phonon replicas. Although both works observe the phonon replicas in the same spectral region, the work by Hwangbo et al.[13] attributes their observations to the coupling between sharp spin–orbit-entangled excitons and the $A_{1g}$ phonon mode at 7.5 THz (253 cm$^{-1}$). Our experimental approach and results, however, identify the origin of these replicas as localized $d$–$d$ transitions, instead of sharp spin–orbit-entangled excitons.

## Methods

**Sample preparation**. We synthesized our NiPS$_3$ crystals using a chemical vapor transport method (for details see ref. [27]). All the powdered elements (purchased from Sigma-Aldrich): nickel (99.99% purity), phosphorus (99.99%), and sulfur (99.998%), were prepared inside an argon-filled glove box. After weighing the starting materials in the correct stoichiometric ratio, we added an additional 5 wt of sulfur to compensate for its high vapor pressure. After the synthesis, we carried out a chemical analysis of the single-crystal samples using a COXI EM-30 scanning electron microscope equipped with a Bruker QUANTAX 70 energy dispersive X-ray system to confirm the correct stoichiometry. We also checked the XRD using a commercial diffractometer (Rigaku Miniflex II). Prior to optical measurements, we determined the crystal axes of the samples using an X-ray diffractometer. We cleaved samples before placing them into high vacuum ($\sim 10^{-7}$ torr) to expose a fresh surface without contamination and oxidation.

**Broadband transient absorption spectroscopy**. 1038 nm (1.19 eV) output of a commercial Yb:KGW regenerative amplifier laser system (PHAROS SP-10-600-PP, Light Conversion) operating at 100 kHz was split into the pump and probe arms. The probe arm was focused onto a calcium fluoride (CaF$_2$) crystal to generate a whitelight continuum, spanning the spectrum from 1.4 to 2.0 eV. Pump arm was used to seed the optical parametric amplifier (ORPHEUS, Light Conversion), which allowed us to tune the wavelength of the pump. Both pump and probe arms were focused onto the sample with an incidence angle of 20 degrees and the beam spot diameters of the pump and probe were 50 and 40 $\mu$m respectively, measured by a knife-edge method. The pump fluence at the sample position is measured to be 4.3 mJ/cm$^2$. The reflected probe beam was sent to a monochromator and a photodiode for a lock-in detection. The spectral resolution of the monochromator is 1 nm. The time resolution of this experiment was near 200 fs. Detailed schematics of the setup are given in Supplementary Information (Supplementary Fig. S1a).

**Energy-resolved coherent phonon spectroscopy**. Ti:sapphire oscillator (Cascade-5, KMLabs) output, centered at 760 nm (1.63 eV) and with pulse duration of $\sim 25$ fs was used. The repetition rate of the laser was set to 80 MHz and no cumulative heating effect was observed. Before splitting the output into pump and probe branches, we precompensated for group velocity dispersion (GVD) using chirp mirrors and N-BK7 wedges to maintain short pulses at the sample position. The pump and probe pulses were characterized separately at the sample position, using the frequency-resolved optical gating technique (Supplementary Fig. S2). The

measured pulse duration was $\sim 25$ fs. To increase the signal-to-noise ratio of our setup, we modulate the pump intensity at 100 kHz. For faster data acquisition and averaging, the pump-probe delay is rapidly scanned at a rate of 5 Hz with an oscillating mirror (APE ScanDelay USB). The reflected probe beam was sent directly to a photodiode for spectrally integrated experiments or was sent to a photodiode through the monochromator for spectrally resolved experiments. The spectral resolution of monochromator is 3 nm. The signal from the photodiode is sent to a lock-in amplifier (Stanford Research SR830) locked to the chopping frequency. Detailed schematics of the setup is given in Supplementary Information (Supplementary Fig. S1b).

## Data availability

The data sets generated and/or analyzed during the current study are available from the corresponding author on reasonable request.

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

## Acknowledgements

We thank Riccardo Comin, Carina Belvin, and Edoardo Baldini for fruitful discussions. We acknowledge support from the US Department of Energy, BES DMSE (data taking and analysis), and Gordon and Betty Moore Foundation's EPiQS Initiative grant GBMF9459 (instrumentation and manuscript writing). H.C.P. was partly supported by Pappalardo Fellowship at MIT. Work at the Center for Quantum Materials was supported by the Leading Researcher Program of the National Research Foundation of Korea (Grant No. 2020R1A3B2079375).

## Author contributions

E.E., B.I., and M.B.Y. built the optical spectroscopy setups. E.E. and B.I. performed the experiments. E.E., B.I., and N.G. analyzed and interpret the data with theoretical input from D.M., H.C.P., and T.S. J.K. synthesized and characterized single crystals of NiPS$_3$, supervised by J.-G.P. E.E., B.I., and N.G. wrote the manuscript with crucial inputs from all authors. This project was supervised by N.G.

## Competing interests

The authors declare no competing interests.
