## [Peer Review File · Nature Communications]

Reviewers' Comments:

Reviewer #1:

Remarks to the Author:

The manuscript by Ergecen et al. reported the observation of phonon replicas through the ultrafast optical spectroscopic study of antiferromagnetic vdW insulator NiPS₃. These phonon replicas only appeared below Neel temperature. So the authors called these states as magnetically brightened dark electron-phonon bound states. Furthermore, the authors employed the energy resolved coherent phonon spectroscopy to differentiate the origin of these electron-phonon bound states. They found that the coupling between localized d-d transitions and A_{1g} phonon mode is more relevant, in comparison to the coupling between spin-orbit-entangled excitons and A_{1g} phonon mode. The results are interesting and may be considered for the publication in Nature Communications.

Here I have some technical questions about the work.

1. There are many phonon modes observed in Raman spectroscopy, as seen in ref. 11. And the A_{1g} phonon mode at ~7 THz or ~225 cm⁻¹ is not so pronounced in Raman spectra. Why is this specific phonon mode coupled to the d-d transitions?

2. The stripy phase of antiferromagnetic order in NiPS₃ is globally inversion symmetric. To understand the correlation between phonon replica and magnetic order, the authors proposed a picture of local inversion symmetry breaking. I feel uneasy about this picture. There may be other possibilities such as magnetic point defects and stacking faults in bulk crystals. I suggest the authors to add their comments in the manuscript.

3. A bulk crystal of NiPS₃ cannot be called 2D magnet. Can the authors show the result in few-layer NiPS₃?

Reviewer #2:

Remarks to the Author:

The manuscript by Ergecen et al., reports the measurements of broadband transient absorption in NiPS₃, revealing a phonon replica state. Further coherent phonon spectroscopy shows a phonon oscillation of the same energy as the phonon replica, which only exists within the spectral energy range that corresponds to d-d excitations. The authors conclude that the d-d excitations are the origin of the phonon replica, and deduce electron-phonon coupling strength. The experimental results, like phonon replica and oscillations in coherent phonon spectrum, are clear and convincing. My biggest scientific concern is the lack of temperature dependence in the coherent phonon spectrum, which seems to contradict the behavior of phonon replica and raise more questions in the authors' data interpretation (explained in my detailed comments). Before addressing this main issue, I cannot recommend its publication yet in Nature Communications.

Below is my detailed comments:

1. Main concern: Temperature dependence

The phonon replica states in the transient absorption spectrum shows the expected temperature dependence, which disappears above T_N, because of the symmetry change across the T_N as proposed by the authors. But in the coherent phonon spectroscopy (Fig. S6), the oscillations, which is attributed to the same phonon mode (~7THz) as in the phonon replica state, is almost identical as a function of temperature. If they correspond to the same phonon mode, what is the reason for this distinctly different temperature dependence?

The author also mentions that this 7 THz mode agrees with previously assigned Raman modes in Ref 11. In Ref 11, the A_g Raman modes of similar energy all show an obvious temperature dependence, which will agree with the observation in transient absorption spectrum but not the coherent phonon spectrum.

Minor comments/concern:

2. Page 3, 2nd line '... with strong vibronic coupling [10]'. I'm not sure if this ref 10 is the best for this purpose.

3. Compared to Fig. 2, it's hard to resolve the phonon replicas in Fig.S4. Is it just because of the

color scaling difference?

4. When extracting the phonon replica energy and oscillation frequency, please also add the error bar the number. It seems to have quite some energy distribution.

5. In Fig. 2b, the arrows indicating energy spacing can be misleading. It should have corresponded to the length start/end at the center of the replica. The size of the arrow (compared to x scale) is also smaller than the extracted 28.5meV.

Reviewer #3:

Remarks to the Author:

Report for “**Magnetically brightened dark electron-phonon bound states in a van der Waals antiferromagnet**”.

In this manuscript, Ergecen and Ilyas *et al* revealed a unique electron-phonon bound state below the Néel transition in NiPS₃, and more importantly, identified the electronic origin as localized d-d excitations and the phonon source as A_{1g} phonon at ~ 7THz, using transient absorption spectroscopy with a dynamic range. Van der Waals (vdW) magnets emerge as a flourish platform that hosts strong coupling between multiple degrees of freedom and realizes novel bound states down to the two-dimension (2D) limit. Yet, it is highly challenging to detect the presence of these bound states and more so to resolve the composition of them. This manuscript has shown, for the first time in vdW magnets, the unambiguous identification of electronic and phononic origins of an electron-phonon bound state with the strongest electron-phonon coupling known so far. This manuscript is of high interest and high quality. I would recommend it for publication at Nature Communications, after addressing the following questions.

1. Could the authors please provide an equilibrium absorption spectrum? How does this equilibrium spectrum compare with the transient absorption spectra? Are the fine oscillations for the electron-phonon coupled states resolvable in the equilibrium spectrum?
2. What is the role of demagnetization with the high intensity pump in detecting the phonon replicas in the transient absorption spectroscopy? Why is the transient reflectivity probed 2 picoseconds after the pump? What is the temporal dynamics of the phonon replicas?
3. When fitting the spectrum with a sum of Gaussians weighted by a Poisson distribution to extract the Huang-Rhys factor g , there are a few questions that I am interested in:
 - a. Why do the authors choose Gaussian, instead of Lorentzian?
 - b. How do the authors determine where the zeroth order of the Poisson distribution is?
 - c. Is there a broad background in the absorption spectrum? What happens if fitting with a broad background plus a sum of Gaussians weighted by a Poisson distribution?
 - d. The linewidth of individual Gaussians should be the convolution between the phonon linewidth and the d-d excitation linewidth. However, the d-d excitation linewidth, reported in PRL **120**, 136402 (2018), seems to be much larger than the Gaussian linewidth here. Any reasons?

A few minor suggestions/comments:

1. There are quite a few “Neel”s in the main text. Perhaps adding the accent in the revision?
2. In “Sample preparation” section, update “NiPS3” to “NiPS₃”.
3. In References, there are a few references missing article or page numbers (e.g., refs. 6-9, 11-21, 23, ...).

Response to Reviewers
(Dated: September 28, 2021)

Reviewer #1 (Remarks to the Author):

The manuscript by Ergecen et al. reported the observation of phonon replicas through the ultrafast optical spectroscopic study of antiferromagnetic vdW insulator NiPS₃. These phonon replicas only appeared below Neel temperature. So the authors called these states as magnetically brightened dark electron-phonon bound states. Furthermore, the authors employed the energy resolved coherent phonon spectroscopy to differentiate the origin of these electron-phonon bound states. They found that the coupling between localized d-d transitions and A_{1g} phonon mode is more relevant, in comparison to the coupling between spin-orbit-entangled excitons and A_{1g} phonon mode. The results are interesting and may be considered for the publication in Nature Communications.

Our response: We thank the reviewer for carefully reading our work and providing insightful remarks, and recommending for publication. Here we try to provide comprehensive answers to the questions.

Here I have some technical questions about the work.

1. There are many phonon modes observed in Raman spectroscopy, as seen in ref. 11. And the A_{1g} phonon mode at ~7 THz or ~225 cm⁻¹ is not so pronounced in Raman spectra. Why is this specific phonon mode coupled to the d-d transitions?

Our response: First, we would like to clarify the phonon mode that strongly couples to the d-d transitions and gives rise to replica formation. As shown in the Supplementary Information, Fig. S5, the Fourier transform of the replica signal is broad and spans an energy interval between 25 meV to 35 meV. This range includes both the A_{1g} phonon mode with 253 cm⁻¹ wavenumber (31 meV) and E_g phonon mode with 225 cm⁻¹ wavenumber (28 meV). Since our transient absorption measurements do not have enough energy resolution to distinguish between these phonons, we perform energy-resolved coherent phonon spectroscopy, which has a better frequency resolution than our transient absorption measurements, to pinpoint the phonons that couple to the d-d transitions.

In NiPS₃, as observed by our coherent phonon spectroscopy measurements, d-d transitions couple to three distinct phonon modes:

- 1) 5.2 THz phonon oscillation, corresponding to an E_g phonon mode with 173 cm⁻¹ wavenumber. This phonon mode has been reported in Raman measurements (Scientific Reports 6, 20904 (2016)).
- 2) 7.5 THz phonon oscillation, corresponding to an A_{1g} phonon mode with 253 cm⁻¹ wavenumber. This phonon mode has been denoted as P5 in Ref. 11, and carries a strong spectral weight as evidenced in Raman measurements. In the manuscript, this phonon mode was referred to as “~7 THz A_{1g} mode”.
- 3) 11.5 THz phonon oscillation, corresponding to an A_{1g} phonon mode with 384 cm⁻¹ wavenumber. This phonon mode has been reported in Raman measurements (Scientific Reports 6, 20904 (2016)).

The E_g phonon mode at 225 cm⁻¹ which has negligible spectral weight in Raman measurements does not appear in our coherent phonon spectroscopy as shown in our Supp. Fig. S7, and therefore has negligible coupling to the d-d transitions. In the light of our coherent spectroscopy measurements, the energy of the A_{1g} phonon mode with 253 cm⁻¹ wavenumber (31 meV) matches the spectral distance between the replicas, as shown in the Fourier transform of the phonon replicas (Supp. Fig. S5).

The energy splitting between the d-d levels is dictated by the overlap between the ligand p-orbitals and d-orbitals. Any modulation that alters this overlap (such as strain, pressure and phonon excitation) will cause a shift in d-d energies. As the reviewer pointed out, in our measurements we only observe phonon replicas formed by the d-d transition and A_{1g} phonon mode with 253 cm⁻¹ wavenumber (31 meV). As shown in Figure 4 of Scientific Reports 6, 20904 (2016), this A_{1g} phonon mode corresponds to the out-of-plane motion of sulfur atoms. This distortion modulates the distance between local nickel sites and sulfur ligands, and therefore leads to replica formation by modulating the d-d transition energy. On the other hand, the other A_{1g} phonon mode with 384 cm⁻¹ wavenumber (11.5 THz) corresponds to the collective out-of-plane motion of nickel and sulfur sites with a slight in-plane component for the sulfur sites. Since this phonon mode does not significantly alter the ligand-transition metal distance, it does not contribute to the replica formation.

We thank the reviewer for insightful comments, and we have made the following changes in the text to clarify the phonon modes that couple to the d-d transition and their signatures in Raman and coherent phonon spectroscopy:

- A) [They manifest themselves... - Line 38]** - We clarified the frequency of the phonon mode that participates in the replica formation.
- B) [The energy spacing between - Line 66]** - We rephrase this sentence to better describe the energy distribution of the replicas seen in our transient absorption measurements.
- C) [As corroborated with ... - Line 76]** - We added a new reference that describes the real space motion of phonons⁸
- D) [This observation indicates ... - Line 77]** - We clarified the frequency of the phonon mode that participates in the replica formation.

- E) **[We observe two ... - Line 94]** - We added a new sentence that summarizes the phonon modes observed in our coherent phonon spectroscopy measurements.
- F) **[The frequency of the dominant ... - Line 95]** - We clarified the frequency of the phonon mode that participates in the replica formation.
- G) **[These spectral features start ... - Line 130]** - We clarified the frequency of the phonon mode that participates in the replica formation.

2. The stripy phase of antiferromagnetic order in NiPS₃ is globally inversion symmetric. To understand the correlation between phonon replica and magnetic order, the authors proposed a picture of local inversion symmetry breaking. I feel uneasy about this picture. There may be other possibilities such as magnetic point defects and stacking faults in bulk crystals. I suggest the authors add their comments in the manuscript.

Our response: Because of dipole transition rules, transitions between d-levels are not allowed if the inversion symmetry of the transition metal atom is not broken. In the case of NiPS₃, the d-d transitions are silent above the magnetic ordering temperature, as the nickel atom is an inversion center. The appearance of d-d transitions at the onset of magnetic order requires a mechanism that links the long range magnetic order and the on-site inversion symmetry breaking. The stacking faults would not give rise to temperature dependent d-d transitions and replica peaks. In addition, magnetic point defects and dislocations can cause loss of inversion symmetry adjacent to the defect sites and can couple to the magnetic order. However, even though defects can influence the optical properties in their vicinity, we think that they cannot give rise to an optical signal that is independent of sample position. In addition, existence of sharp magnetic excitons in our samples is also indicative of high sample quality and sparse defect distribution.

Following the reviewer's comments, we have added the following comments in the revised text, stating that magnetic point defects and stacking faults are unlikely to result in temperature dependent phonon replicas of d-d transitions:

- 1) **[Although stacking faults and lattice defects... - Line 138]** - We added a sentence to discuss the stacking faults and lattice defects that can give rise to inversion symmetry breaking.

3. A bulk crystal of NiPS₃ cannot be called 2D magnet. Can the authors show the result in few-layer NiPS₃?

Our response: We agree with the reviewer on the fact that a bulk crystal of NiPS₃ cannot be called a 2D magnet. However, the phenomenon reported here for bulk crystals should be valid for few layer flakes as long as the magnetic order is preserved. This is due to the fact that d-d transitions are localized transitions, and their properties do not depend on the interlayer coupling. As the magnetic order gets suppressed in the monolayer limit (Ref. 11), the phonon replicas are not expected in the monolayer limit. In addition, phonon replicas have been shown to be existent

down to bilayer limit in another publication (see Nature Nanotechnology 16, 655-660, 2021) published during the preparation of this manuscript.

Reviewer #2 (Remarks to the Author):

The manuscript by Ergecen etc., reports the measurements of broadband transient absorption in NiPS₃, revealing a phonon replica state. Further coherent phonon spectroscopy shows a phonon oscillation of the same energy as the phonon replica, which only exists within the spectral energy range that corresponds to d-d excitations. The authors conclude that the d-d excitations as the origin of the phonon replica, and deduce electron-phonon coupling strength. The experimental results, like phonon replica and oscillations in coherent phonon spectrum, are clear and convincing. My biggest scientific concern is the lack of temperature dependence in the coherent phonon spectrum, which seems to contradict the behavior of phonon replica and raise more questions in the authors' data interpretation (explained in my detailed comments). Before addressing this main issue, I cannot recommend its publication yet in Nature Communications. Below is my detailed comments:

1. Main concern: Temperature dependence

The phonon replica states in the transient absorption spectrum shows the expected temperature dependence, which disappears above T_N, because of the symmetry change across the T_N as proposed by the authors. But in the coherent phonon spectroscopy (Fig. S6), the oscillations, which is attributed to the same phonon mode (~7THz) as in the phonon replica state, is almost identical as a function of temperature. If they correspond to the same phonon mode, what is the reason for this distinctly different temperature dependence?

The author also mentions that this 7 THz mode agrees with previously assigned Raman modes in Ref 11. In Ref 11, the A_g Raman modes of similar energy all show an obvious temperature dependence, which will agree with the observation in transient absorption spectrum but not the coherent phonon spectrum.

Our response: We thank the referee for examining our work thoroughly and providing insightful comments and inputs.

First, we would like to point out that the Fourier transform of the replica signal is broad and spans an energy interval between 25 meV to 35 meV (Supp. Fig. S5). This range includes both the A_{1g} phonon mode with 253 cm⁻¹ wavenumber (31 meV) and E_g phonon mode with 225 cm⁻¹ wavenumber (28 meV). To pinpoint the phonon mode that is responsible for replica formation, we perform energy-resolved coherent phonon spectroscopy. The most pronounced phonon mode observed in our coherent phonon spectroscopy is the A_{1g} phonon mode with 253 cm⁻¹ wavenumber. In the Raman spectrum, this corresponds to the mode denoted as P5 in Ref. 11. The energy of the A_{1g} phonon mode with 253 cm⁻¹ wavenumber (31 meV) matches the spectral distance between the replicas. On the other hand, the phonon mode at 225 cm⁻¹ (denoted as P4 in Ref. 11), which has negligible spectral weight in Raman measurements, does not appear in our

coherent phonon spectroscopy, as shown in Supp. Fig. S7. Therefore, the phonon mode at 225 cm^{-1} has negligible coupling to the d-d transitions and cannot be responsible for replica formation.

We would like to point out that the temperature dependence of both P4 and P5 phonon amplitudes in Ref. 11, is not correlated with the magnetic ordering temperature. Furthermore, both of them exist above and below the magnetic ordering temperature. This observation indicates that the Raman amplitude for 253 cm^{-1} phonon mode is not directly influenced by the magnetic order. As we have mentioned in the main text, the coupling between the d-d transitions and 253 cm^{-1} phonon mode exists above the magnetic ordering temperature, but they are not optically active because of local inversion symmetry. At low temperatures, d-d transitions become optically active because of local inversion symmetry breaking arising from the magnetic order.

As the referee alluded to, the Raman amplitudes of both P4 (225 cm^{-1}) and P5 (253 cm^{-1}) phonon modes show temperature dependence, and this dependence is not observed in our coherent phonon spectroscopy data. Even though both Raman and coherent phonon spectroscopy are sensitive to phonon modes, the excitation and detection of the phonon modes are completely different for these spectroscopy modalities. In Raman, a single photon inelastically scatters and spontaneously creates a single phonon excitation through a virtual transition, highly detuned from equilibrium excited states. Thus, the Raman amplitude depends on the structure of the equilibrium excited state (detuning etc.) and the thermal occupation of phonon modes, which can change as a function of temperature.

On the other hand, in coherent phonon spectroscopy, a probe pulse samples the real time phase coherent phonon oscillations following a pump excitation. Unlike Raman scattering, these coherent phonon oscillations are launched by impulsive Raman scattering and displacive excitation of coherent phonons (DECP). In the case of DECP, the amplitude of the phonon mode is proportional to the phonon displacement induced by the pump pulse, which is determined by the difference between equilibrium and nonequilibrium lattice positions. Our coherent phonon spectroscopy data for NiPS₃ implies that the excitation amplitude of the P5 phonon mode does not change with temperature. We think that the discrepancy in temperature dependences of the Raman and coherent phonon spectroscopy is not unexpected, as they can launch and probe phonon modes differently.

To further clarify the differences between Raman and coherent phonon spectroscopy, we would like to compare and contrast two studies (one Raman and one coherent phonon spectroscopy) performed on isostructural & isoelectronic van der Waals magnets CrSiTe₃ & CrGeTe₃. For these compounds, the A_{1g} phonon mode at 136 cm^{-1} does not show any temperature dependence in Raman (Fig. 4 - <https://arxiv.org/pdf/1604.08745.pdf>), whereas coherent phonon spectroscopy (<https://arxiv.org/pdf/1910.06376.pdf>) shows a dramatic change in amplitude because of a spin dependent phonon excitation mechanism. We hope that this example clears up the differences between coherent phonon spectroscopy and Raman measurements.

We have added a supplementary note (Supp. Note 10) that describes the differences between Raman and coherent phonon spectroscopy more clearly, and made the following changes in the main text:

- 1) **[They manifest themselves... - Line 38]** - We clarified the frequency of the phonon mode that participates in the replica formation.
- 2) **[The energy spacing between - Line 66]** - We rephrase this sentence to better describe the energy distribution of the replicas seen in our transient absorption measurements.
- 3) **[As corroborated with ... - Line 76]** - We added a new reference that describes the real space motion of phonons.
- 4) **[This observation indicates ... - Line 79]** - We clarified the frequency of the phonon mode that participates in the replica formation.
- 5) **[Using the broadband ... - Line 89]** - We added the phrase “displacive” to clarify the mechanisms that lead to ultrafast phonon excitation. We also mention that both A_{1g} and E_g symmetry modes can be excited through coherent phonon spectroscopy.
- 6) **[We observe two ... - Line 94]** - We added a new sentence that summarizes the phonon modes observed in our coherent phonon spectroscopy measurements.
- 7) **[The frequency of the dominant ... - Line 95]** - We clarified the frequency of the phonon mode that participates in the replica formation.
- 8) **[These spectral features start ... - Line 130]** - We clarified the frequency of the phonon mode that participates in the replica formation.

Minor comments/concern:

2. Page 3, 2nd line '... with strong vibronic coupling [10]'. I'm not sure if this ref 10 is the best for this purpose.

Our response: This was a mistake. We have corrected it by replacing the reference with the right one. We thank the referee for pointing this out.

3. Compared to Fig. 2, it's hard to resolve the phonon replicas in Fig.S4. Is it just because of the color scaling difference?

Our response: We agree with the referee. In Fig. S4, it is harder to resolve the phonon replicas. The data in Fig. 2, was taken by fixing the delay time (at 2 ps), and averaging for a longer time. On the other hand, the data in Fig. S4 had been averaged less for the sake of time, since we are examining the time dependence as well.

4. When extracting the phonon replica energy and oscillation frequency, please also add the error bar the number. It seems to have quite some energy distribution.

Our response: We thank the referee for bringing this to our attention. Following this comment, we made the following changes:

- 1) **[Supp. Note 9]** - we have added the error bars to the extracted values of replica energy and oscillation frequency for two different undressed d-d transition lineshapes, Gaussian and Lorentzian.
- 2) **[The energy spacing between - Line 66]** - We rephrase this sentence to better describe the energy distribution of the replicas seen in our transient absorption measurements.

5. In Fig. 2b, the arrows indicating energy spacing can be misleading. It should have corresponded to the length start/end at the center of the replica. The size of the arrow (compared to x scale) is also smaller than the extracted 28.5meV.

Our response: We have corrected this in our revised version of the manuscript. The arrows were for demonstration purposes only. We removed them in our revised manuscript. Thanks for pointing this out.

Reviewer #3 (Remarks to the Author):

Report for “**Magnetically brightened dark electron-phonon bound states in a van der Waals antiferromagnet**”.

In this manuscript, Ergecen and Ilyas *et al* revealed a unique electron-phonon bound state below the Néel transition in NiPS₃, and more importantly, identified the electronic origin as localized d-d excitations and the phonon source as A_{1g} phonon at ~ 7THz, using transient absorption spectroscopy with a dynamic range. Van der Waals (vdW) magnets emerge as a flourish platform that hosts strong coupling between multiple degrees of freedom and realizes novel bound states down to the two-dimension (2D) limit. Yet, it is highly challenging to detect the presence of these bound states and more so to resolve the composition of them. This manuscript has shown, for the first time in vdW magnets, the unambiguous identification of electronic and phononic origins of an electron-phonon bound state with the strongest electron-phonon coupling known so far. This manuscript is of high interest and high quality. I would recommend it for publication at Nature Communications, after addressing the following questions.

1. Could the authors please provide an equilibrium absorption spectrum? How does this equilibrium spectrum compare with the transient absorption spectra? Are the fine oscillations for the electron phonon coupled states resolvable in the equilibrium spectrum?

Our response: We are thankful to the referee for carefully reading our work, and appreciating the importance of our findings.

The equilibrium absorption spectrum of NiPS₃ has been reported in a recent publication (Phys. Rev. Lett. 120, 136402 (2018) - Fig. S6). The equilibrium absorption spectrum cannot resolve the fine spectral oscillations. This fact highlights the power of transient absorption measurements, which have a high dynamic range and allow us to observe faint spectral details.

In addition, equilibrium linear dichroism (birefringence) spectroscopy measurements have also observed the phonon replicas (see Nature Nanotechnology 16, 655-660, 2021) published during the preparation of this manuscript. Because NiPS₃ exhibits magnetically induced birefringence, equilibrium linear dichroism (birefringence) spectroscopy is sensitive to the magnetic order induced spectral features with high dynamic range.

2. What is the role of demagnetization with the high intensity pump in detecting the phonon replicas in the transient absorption spectroscopy? Why is the transient reflectivity probed 2 picoseconds after the pump? What is the temporal dynamics of the phonon replicas?

Our response: The high intensity pump pulse heats up the electronic system by efficiently generating electron-hole pairs. This in turn melts the magnetic order and “washes out” any spectral features pertinent to magnetic order. We measure the difference between this nonequilibrium absorption spectrum (at 2 ps time delay) and equilibrium one (at negative time delay). This is what allows us to measure small spectral signals, which cannot be detected with conventional equilibrium absorption methods.

The 2 ps time delay is not a special point. We actually have examined the temporal dynamics of phonon replicas up to 100 ps delay times (see Fig. S4) and have observed that phonon replicas survive up to this delay times with no visible change in oscillation amplitudes. This indicates that the spectral oscillations are indeed equilibrium phenomena.

3. When fitting the spectrum with a sum of Gaussians weighted by a Poisson distribution to extract the Huang-Rhys factor g , there are a few questions that I am interested in:
 - a. Why do the authors choose Gaussian, instead of Lorentzian?

Our response: The model we use to obtain the Huang-Rhys factor g takes the undressed d-d transition energy as a free parameter. In our fitting procedures, we both used Gaussian and Lorentzian distributions for the undressed d-d transition lineshapes. The selection of the lineshape function does not significantly change the extracted Huang-Rhys factor and the phonon frequency. Therefore, none of the conclusions are affected by the selection of the lineshape function.

We have added a subsection into Supp. Note 9 that shows the fitting results obtained using a Lorentzian lineshape.

- b. How do the authors determine where the zeroth order of the Poisson distribution is?

Our response: In our fits, the model we use takes the bare d-d transition energy as a free parameter. The fit outputs the zeroth order of the Poisson distribution.

- c. Is there a broad background in the absorption spectrum? What happens if fitting with a broad background plus a sum of Gaussians weighted by a Poisson distribution?

Our response: The model used in this paper fits the d-d transition region very well without any need of a background.

- d. The linewidth of individual Gaussians should be the convolution between the phonon linewidth and the d-d excitation linewidth. However, the d-d excitation linewidth, reported in PRL 120, 136402 (2018), seems to be much larger than the Gaussian linewidth here. Any reasons?

Our response: The d-d excitation linewidth as reported in PRL 120, 136402 (2018) is larger than the Gaussian linewidth of bare d-d transitions. The detection method used in PRL 120, 136402 (2018) was not capable of resolving the phonon replicas, and therefore the paper reports the linewidth of the phonon replicas as the linewidth of bare d-d transitions. There are actually replica bands lying under the d-d excitation lineshape. Typically, the d-d lines are fairly sharp and they mainly get broadened due to strong electron-phonon interactions. We claim that what has been shown in PRL 120, 136402 (2018), is actually d-d transition with many phonon sidebands, which were not resolved in that particular experimental approach.

A few minor suggestions/comments:

1. There are quite a few “Neel”s in the main text. Perhaps adding the accent in the revision?

Our response: We thank the reviewer for comments that will definitely improve our manuscript. We have made changes in the spelling.

2. In “Sample preparation” section, update “NiPS3” to “NiPS₃”.

Our response: We have corrected this.

3. In References, there are a few references missing article or page numbers (e.g., refs. 6-9, 11-21, 23, ...).

Our response: We have corrected this.

Summary of Changes

In this document, we summarized the revisions of the main manuscript and Supplementary Information file of the article “Magnetically brightened dark electron-phonon bound states in a van der Waals antiferromagnet”. Edits are highlighted in red in the revised documents. When applicable, we also reference the reviewer’s comment addressed by a given item. Square brackets enclose the beginning of each section and line numbers:

- 1) [... to the lattice... - Line 26] - We replaced the word “modes” with “states”
- 2) [... proxy for antimagnetic order - Line 27] - We replaced the phrase “localized d-orbitals” with “d-d transition in transition metals”
- 3) [This requires spectral tools that can... - Line 34] - We added “partaking in the bound state formation” to better explain the motivation for using various spectral tools to understand the constituents of replica bands/bound states.
- 4) [They manifest themselves... - Line 40] - We clarified the frequency of the phonon mode that participates in the replica formation.
- 5) [The energy spacing between - Line 66] - We rephrase this sentence to better describe the energy distribution of the replicas seen in our transient absorption measurements.
- 6) [As corroborated with ... - Line 76] - We added a new reference that describes the real space motion of phonons.
- 7) [This observation indicates ... - Line 78] - We clarified the frequency of the phonon mode that participates in the replica formation.
- 8) [... scheme a low intensity ... - Line 89] - We added the phrase “or displacively” to clarify the mechanisms that lead to ultrafast phonon excitation. We also mention that both A_{1g} and E_g symmetry modes can be excited through coherent phonon spectroscopy.
- 9) [Subsequent to ... - Line 90] - We removed “A_{1g}” and used “these”.
- 10) [We observe two ... - Line 94] - We added a new sentence that summarizes the phonon modes observed in our coherent phonon spectroscopy measurements.
- 11) [The frequency of the dominant ... - Line 95] - We clarified the frequency of the phonon mode that participates in the replica formation.
- 12) [...ion and the ligands ... - Line 113] - We added “7.5 THz”
- 13) [...operators of the ... - Line 117] - We added “7.5 THz”
- 14) [where L is... - Line 123] - We removed “Gaussian”.
- 15) [These spectral features start ... - Line 129] - We clarified the frequency of the phonon mode that participates in the replica formation.
- 16) [Although transitions between.. Line 136] - We replaced the word “Although” with “Generally”
- 17) [Although stacking faults and lattice defects... - Line 138] - We added a sentence to discuss the stacking faults and lattice defects that can give rise to inversion symmetry breaking.
- 18) [Although both works observe... - Line 174] - We clarified the frequency of the phonon mode that participates in the replica formation.

19) [The spectral resolution... - Line 196] - We added a line to describe the spectral resolution of our transient reflectivity spectroscopy setup.

In the figures and captions given in the main text, we made following changes:

- 1) **[Figure 1]** - We added frequency value of A_{1g} mode (7.5 THz), that is involved in replica formation.
- 2) **[Figure 2]** - We removed the arrows that had been used to delineate the separation between replica peaks.
- 3) **[Figure 2 Caption]** - We removed the sentence that described the separation between replica peaks.
- 4) **[Figure 3a]** - We added the labels for other phonon modes observed in coherent phonon spectroscopy.
- 5) **[Figure 3a Caption]** - We added the exact phonon frequencies to the inset.
- 6) **[Figure 4 Caption]** - We replaced Gaussian with the linewidth function.

In the Supplementary Information, we made following changes in each section:

- 1) **[Supplementary Note 9 - In the minimal model...]** - We added our fitting results that are obtained using a Lorentzian function for undressed d-d transitions instead of a Gaussian one.
- 2) **[Supplementary Note 10]** - We added a supplementary note that explains the differences between Raman scattering and coherent phonon spectroscopy.

Reviewers' Comments:

Reviewer #1:

Remarks to the Author:

The authors have properly addressed my questions and concerns. I now support the publication of this work in Nature Communications.

Reviewer #2:

Remarks to the Author:

The authors have addressed my concerns. The extended discussion on the difference between coherent photon spectroscopy and Raman is very helpful and clarifies potential issues in the manuscript. I can now recommend the publication in Nature Communications.

Reviewer #3:

Remarks to the Author:

The authors have addressed all my questions satisfactorily. I have no further questions and would be happy to recommend publication of this work at Nature Communications.

REVIEWERS' COMMENTS

Reviewer #1 (Remarks to the Author):

The authors have properly addressed my questions and concerns. I now support the publication of this work in Nature Communications.

Our response: We thank the reviewer for carefully reading our work and recommending for publication.

Reviewer #2 (Remarks to the Author):

The authors have addressed my concerns. The extended discussion on the difference between coherent photon spectroscopy and Raman is very helpful and clarifies potential issues in the manuscript. I can now recommend the publication in Nature Communications.

Our response: We thank the reviewer for taking the time to assess our work and supporting it for publication.

Reviewer #3 (Remarks to the Author):

The authors have addressed all my questions satisfactorily. I have no further questions and would be happy to recommend publication of this work at Nature Communications.

Our response: We thank the reviewer for reviewing our work and recommending for publication.